# Identification and Potential Participation of Lipases in Autophagic Body Degradation in Embryonic Axes of Lupin (*Lupinus* spp.) Germinating Seeds

**DOI:** 10.3390/ijms25010090

**Published:** 2023-12-20

**Authors:** Karolina Wleklik, Szymon Stefaniak, Katarzyna Nuc, Małgorzata Pietrowska-Borek, Sławomir Borek

**Affiliations:** 1Department of Plant Physiology, Faculty of Biology, Adam Mickiewicz University Poznań, Uniwersytetu Poznańskiego 6, 61-614 Poznań, Poland; karolina.wleklik@amu.edu.pl (K.W.); szymon.stefaniak@amu.edu.pl (S.S.); 2Department of Biochemistry and Biotechnology, Faculty of Agronomy, Horticulture and Bioengineering, Poznań University of Life Sciences, Dojazd 11, 60-632 Poznań, Poland; katarzyna.nuc@up.poznan.pl (K.N.); malgorzata.pietrowska-borek@up.poznan.pl (M.P.-B.)

**Keywords:** asparagine, Atg15, autophagy, carbon starvation, lupin, seed embryo, sucrose, transcriptomics, vacuole, yeast

## Abstract

Autophagy is a fundamental process for plants that plays a crucial role in maintaining cellular homeostasis and promoting survival in response to various environmental stresses. One of the lesser-known stages of plant autophagy is the degradation of autophagic bodies in vacuoles. To this day, no plant vacuolar enzyme has been confirmed to be involved in this process. On the other hand, several enzymes have been described in yeast (*Saccharomyces cerevisiae*), including Atg15, that possess lipolytic activity. In this preliminary study, which was conducted on isolated embryonic axes of the white lupin (*Lupinus albus* L.) and Andean lupin (*Lupinus mutabilis* Sweet), the potential involvement of plant vacuolar lipases in the degradation of autophagic bodies was investigated. We identified in transcriptomes (using next-generation sequencing (NGS)) of white and Andean lupin embryonic axes 38 lipases with predicted vacuolar localization, and for three of them, similarities in amino acid sequences with yeast Atg15 were found. A comparative transcriptome analysis of lupin isolated embryonic axes cultured in vitro under different sucrose and asparagine nutrition, evaluating the relations in the levels of the transcripts of lipase genes, was also carried out. A clear decrease in lipase gene transcript levels caused by asparagine, a key amino acid in lupin seed metabolism which retards the degradation of autophagic bodies during sugar-starvation-induced autophagy in lupin embryonic axes, was detected. Although the question of whether lipases are involved in the degradation of autophagic bodies during plant autophagy is still open, our findings strongly support such a hypothesis.

## 1. Introduction

Autophagy is a bulk degradation system that occurs similarly in fungal, animal, and plant cells. It plays a crucial role in the auto-digestion of cell components to degrade damaged or toxic components and recycle nutrients [1]. After several decades of research on autophagy in plants, what was once perceived as a simple, non-selective process of degradation of cell components is now understood to be a highly sophisticated and selective process involved in virtually all aspects of plant physiology. In plants, the following two major autophagic pathways have been described: macroautophagy and microautophagy [2]. Macroautophagy is initiated in the cytoplasm with the formation of cup-shaped membranes, called phagophores, that enclose the cargo to be degraded. The final stage of phagophore differentiation is the complete surrounding of the cargo and its sequestration inside an autophagosome. This is a vesicle with a double, bilayer lipid-protein membrane containing cargo intended for autophagic degradation. In plants and yeasts, autophagosomes fuse with vacuoles to create autophagic bodies [2,3,4]. During microautophagy, cargo is engulfed directly by a vacuole through the invagination of a tonoplast, followed by pinching off of the membrane to release a vesicle containing the cytoplasmic constituents inside the vacuole lumen [5]. Degradation of autophagic bodies occurs rapidly and begins immediately after their appearance in a vacuole [6]. Only a few publications have described this stage of plant autophagy, and, in addition, such descriptions are usually vague, not providing details about the cytological and molecular mechanisms involved [2,3,7]. Compared to the initial stages of autophagy, the mechanisms and regulation of the degradation of autophagic bodies and the efflux of metabolites from a vacuole to the cytoplasm are very poorly investigated and understood. However, these are key stages on the path to recycling cellular components in the entire autophagy process.

It should be emphasized that it is still unknown which vacuolar enzymes are required for autophagic body degradation in plants. In contrast, enzymes required for autophagic body breakdown in yeasts have been known since the early 1990s [8], showing how neglected this aspect of plant autophagy is. Two of the important uncertainties are the existence of a functional plant analog of the yeast vacuolar putative lipase Atg15/Aut5/Cvt17 and its involvement in the final steps of autophagy, in particular, the degradation of autophagic bodies [9,10]. Atg15 is the best-known and best-described protein involved in autophagic body breakdown in yeast [11,12,13,14]. Determining this unknown area of the involvement of particular proteins and their roles in the degradation of autophagic bodies in plants would be a major achievement on the road to a complete understanding of the course of autophagy in these organisms.

Carbon starvation is a well-known stressor enhancing autophagy in plant cells [15,16,17,18]. The intensification of autophagic degradation provides cells with respiratory substrates, which simultaneously enhances their survival potential [19]. Our previous work indicated that autophagy occurred in the cells of isolated embryonic axes of germinating lupin (*Lupinus* spp.) seeds cultured in vitro on a mineral medium for 96 h under carbon (sugar) starvation conditions (−S). The presence of autophagy was reflected in the huge cell vacuolization [18], decreased content of phosphatidylcholine [18], and higher level of *atg8* gene transcripts [18]. We also induced a slowdown in the degradation of autophagic bodies by simultaneously nourishing the lupin embryonic axes with asparagine (−S+Asn), a central amino acid in the metabolism of lupin seeds [15,20]. In the cells of the −S+Asn lupin embryonic axes, we observed an accumulation of autophagic bodies in the vacuoles. Such ultrastructures are unusual in sugar-starved axes (−S) as the process of autophagic body breakdown is rapid and vacuoles rather quickly become clear [16]. In addition, visualization of the autophagic body degradation in yeast cells was not easy to achieve, even with the use of live-cell microscopy [21]. This suggested that asparagine in plants has an inhibitory effect on this late step of autophagy. Apart from the accumulation of autophagic bodies, it was also noted that in the −S+Asn axes, the lipolytic activity was significantly decreased in comparison to the −S axes [16,18]. The action of exogenous asparagine, which slowed down the degradation of autophagic bodies and decreased lipolytic activity in the −S+Asn axes, convinced us to use this amino acid in our research concerning the role of vacuolar lipases in autophagic body degradation. The comparison of our previously obtained results [15,16,18] with the currently performed transcriptome analysis (NGS and SRA databases, BioProject accession number for white lupin: PRJNA953600, and Andean lupin: PRJNA953433, deposited in 2023) suggested that vacuolar lipases have a high potential to be involved in the degradation of autophagic bodies in plants. This preliminary study provides new insights into one of the final, poorly understood stages of autophagy in plants, which is the degradation of autophagic bodies.

## 2. Results

Research was performed on the embryonic axes isolated from the imbibed seeds of the white lupin (*Lupinus albus* L.) and Andean lupin (*Lupinus mutabilis* Sweet) and cultured in vitro for 96 h on a liquid mineral Heller medium [22] with or without 60 mM of sucrose (+S and −S, respectively) and with 35 mM of asparagine (−S+Asn and +S+Asn, respectively). A detailed description of the morpho-physiological parameters of the 96 h lupin embryonic axes was published previously [16,18]. Based on the results of the mass sequencing of the transcriptomes of the white and Andean lupin isolated embryonic axes cultured in vitro for 96 h, we identified 90 and 111 transcripts of the genes encoding lipases for the embryonic axes of the white lupin and Andean lupin, respectively (Appendix A). The identified genes encoding lipases were analyzed in terms of their predicted subcellular localization, and the full list is presented in Appendix A. Among all of them, we found that 38 lipases demonstrated probable vacuolar localization (Table 1 and Appendix A). Since the degradation of autophagic bodies occurs only in vacuoles and yeast Atg15 is also the only lipase localized in vacuoles [11,12], we decided to exclude lipases with other cell localizations from our study.

The amino acid sequence of the Atg15 (QHB07198.1_SC) yeast with 38 amino acid sequences of lupin lipases with predicted vacuolar localization was compared (Table 1). As reported previously, the Atg15 yeast possesses lipolytic activity and is involved in autophagic body degradation [11,12,13,14,23,24,25]. Our analysis revealed that three lupin lipases (XP_019430451.1, XP_019425758.1, and XP_019420123.1) had high amino acid sequence similarity to the Atg15 yeast (Figure 1). The degree of identity and similarity for the individual proteins to Atg15 was as follows: XP_019430451.1—16.5% and 32%, respectively; XP_019425758.1—18.6% and 28.5%, respectively; and XP_019420123.1—17.8% and 32%, respectively. The transcript of monoacylglycerol lipase-like protein (XP_019420123.1) was detected only in Andean lupin, whereas the other two lipases were present in both the white and Andean lupin species. These three lupin lipases with the highest amino acid sequence similarity to Atg15 are marked with red font in Table 1.

The lupin lipases (XP_019420123.1, XP_019425758.1, and XP_019430451.1) most related to Atg15 (QHB07198.1; Figure 1) were found to have the IWVTGHSLGG amino acid sequence, which is a putative lipase active-site motif of the Atg15 yeast [11]. The amino acid sequence of the selected lupin lipases was entered into MegAlign Pro (DNASTAR Lasergene) and aligned using the Clustal Omega algorithm. The global sequence alignment (Figure 2) revealed that XP_019420123.1 was 17.8% identical and 32% similar to Atg15 while XP_019425758.1 was 18% identical and 31% similar, and XP_019430451.1 was 16.5% identical and 32% similar to Atg15. However, the analysis of the presence of a putative lipase active-site motif revealed that the amino acid sequence in this region was highly conservative, and for both XP_019420123.1 and XP_019425758.1, the identity was 80% and similarity was 90%, while for XP_019430451.1, these values were much lower (40% and 50%, respectively). The amino acid sequences in this region were identical in both XP_019425758.1 and XP_019420123.1 (Figure 2). It is worth adding here that both XP_019420123.1 and XP_019425758.1 had high probabilities of vacuolar localization. XP_019420123.1 was first (with the highest score of 0.7413) and XP_019425758.1 was fourth (with a score of 0.7070) on the lupin lipase list with potential vacuolar localization (Table 1).

Changes in the levels of the gene transcripts encoding probable vacuolar lipases in the lupin embryonic axes obtained from all four trophic variants of the in vitro culture (+S, +S+Asn, −S, and −S+Asn) were also analyzed, and we visualized them in the form of heat maps (Figure 3). The picture of the changes was more unequivocal for white lupin. It showed decreased levels of most of the transcripts of the genes encoding probable vacuolar lipases under sugar-starvation conditions (−S) in comparison to the sucrose-fed axes (+S and +S+Asn), and these decrease were even more evident in the sugar-starved and asparagine-fed (−S+Asn) axes. In Andean lupin, the most visible change was decreased levels of transcripts encoding the genes of the probable vacuolar lipases in the axes fed with asparagine (+S+Asn and −S+Asn) compared (in pairs) to the non-asparagine-fed axes (+S and −S). 

## 3. Discussion

The Atg15 yeast possesses lipolytic activity and is one of the enzymes whose involvement in the degradation of the autophagic body has been confirmed [11,12,13,14,23,24,25]. *Saccharomyces cerevisiae* yeast lacking Atg15 accumulated autophagic bodies in vacuoles, demonstrating its essential role in autophagic degradation [11,12]. Such accumulation of autophagic bodies in yeast vacuoles can also be caused by proteinase A (PEP4) or proteinase B (PRB1) deficiencies [8]. Orthologs of Atg15 are not found in organisms other than fungi [11]. However, the degradation of cargo located inside an autophagic body requires earlier disintegration of its phospholipid-built membrane [10]. Therefore, enzymes with lipase or phospholipase activity must initiate their breakdown [13,14,26]. It has been reported that the phospholipase A2 (LPLA-2) is required for the successful processing of autophagic degradation in *Caenorhabditis elegans* [27]. Mutants lacking this enzyme accumulate membrane material in enlarged lysosomes. As the integrity of these membranes is maintained and, thus, cargo release is disrupted, the nutrient deficiency status deepens, resulting in embryonic lethality [27]. The suggestions from the literature together with the results of our previous studies [16,18] and this preliminary research have indicated that some lipases may also be involved in the degradation of autophagic bodies in plants. However, some aspects should still be considered. There is an important concern about Atg15 substrate specification. Atg15 was found to disturb the membranes of vesicles delivered by both multivesicular bodies (MVBs) and autophagy pathways [13,28]. Nevertheless, it is not known whether Atg15 specifies the disintegration of membranes belonging only to autophagic bodies and multivesicular bodies or if it disintegrates other membranes as well. A vacuole, where autophagic body degradation occurs, is surrounded by a single membrane—the tonoplast. Alternatively, Atg15 also causes its disintegration, but this event is followed by the quick recovery of the tonoplast [13]. On the other hand, Atg15 is the only lipase-like protein in yeast vacuoles [13], whereas our results predicted above 30 lipases to be located in lupin vacuoles (Table 1). Therefore, the question arises whether and which plant vacuolar lipases are involved in the breakdown of autophagic bodies. At the current stage of the research, it is not possible to provide a definite answer, but by the evolutionary conservation of the autophagy process among eukaryotes, it can be supposed that the best candidates would be lipases with similar amino acid sequences of active sites to the Atg15 yeast.

Our transcriptome analysis identified 90 and 111 transcripts of genes encoding lipases in the embryonic axes of white lupin and Andean lupin, respectively (Appendix A). Among these transcripts, 38 lipases represented probable vacuolar localization (Table 1). In our research, the transcript levels of the genes encoding lipases with predicted vacuolar localization (Figure 3) did not increase under sugar-starvation conditions (−S), but we previously recorded significantly elevated lipolytic activity under these trophic conditions [15,16,18] and discordantly high total lipid contents in comparison to the sucrose-fed axes (+S) [15,16,18]. This may suggest that elevated lipolytic activity in −S lupin axes may be caused only by selected lipases that likely are not involved in the decomposition of storage lipids accumulated in lipid droplets deposited in the cytoplasm. Therefore, we postulated that a significant part of this increased lipolytic activity originated from vacuoles because, simultaneously with the elevated total lipolytic activity, we observed an unexpectedly high level of total lipids [15,16,18]. During seed germination and embryo growth, the degradation of storage lipids is initiated by lipases operating on lipid droplets located in the cytoplasm [15,29]. Therefore, the source of the elevated total lipolytic activity observed in the lupin embryonic axes under sugar starvation conditions (−S) could not be only the cytoplasm and the processes related to lipid-droplet degradation. There must have been another source responsible for the elevated lipolytic activity in the −S axes, and we pointed to vacuolar lipases in this context. It must be added here that asparagine caused a clear decrease in a majority of the vacuolar lipase transcripts, especially in the white lupin axes (Figure 3). Asparagine also significantly decreased the total lipolytic activity [16,18] and significantly retarded the degradation of the autophagic bodies in the −S+Asn lupin axes [16,18]. Thus, taking these data together, we hypothesized that vacuolar lipases (at least some of them) participated in the degradation of autophagic bodies in plants.

We also paid attention to the changes in the levels of the transcripts of the genes encoding lipases with indicated similarity to Atg15 (marked with asterisks and red font in Figure 3). In almost all these cases, the levels of the transcripts were elevated in the sugar-starved axes (−S) in comparison to the sucrose-fed axes (+S). Moreover, two lipases found in both of the lupin species (XP_019430451.1 and XP_019425758.1) showed relatively high probabilities for vacuolar localization (marked with red font in Table 1). These results may suggest that specific vacuolar lipases are responsible for the elevation of lipolytic activity in sugar-starved axes (−S) [15,16,18], and they may also be involved in the degradation of autophagic bodies during sugar-starvation-induced autophagy.

## 4. Materials and Methods

### 4.1. Plant Material

Seeds of white lupin (*Lupinus albus* L.) and Andean lupin (*Lupinus mutabilis* Sweet) were sterilized in 0.02% HgCl_2_ for up to 20 min and then thoroughly washed several times with autoclaved distilled water. Embryonic axes were isolated from the sterilized seeds imbibed in the dark for 24 h at 25 °C. The axes were placed on autoclaved filter paper (Whatman no. 3) in sterile tubes on liquid mineral Heller medium [22] under the following four trophic conditions: with 60 mM of sucrose (+S), with 60 mM of sucrose and 35 mM of asparagine (+S+Asn), without sucrose (−S), and without sucrose and with 35 mM of asparagine (−S+Asn). These media were autoclaved, except for the asparagine-containing media, which were sterilized using 0.22 μm Millipore filters. The isolated embryonic axes were cultured in vitro in the dark for 96 h at 25 °C.

### 4.2. Transcriptomics—NGS

White and Andean lupin embryonic axes were isolated from the imbibed seeds and cultured in vitro for 96 h under the trophic conditions described above (+S, −S, +S+Asn, and −S+Asn) and frozen in liquid nitrogen and stored at −80 °C until RNA isolation. The frozen embryonic axes were powdered in liquid nitrogen and the total RNA was isolated from the samples using an RNeasy Plant Mini Kit (Qiagen, Hilden, Germany). All libraries were prepared as described in our previous paper [18]. The quantity of the indexed libraries was estimated by a Qubit fluorimeter (Thermo Fisher Scientific, Waltham, MA, USA), mixed in equal amounts, and sent for NGS to Fasteris (Life Science, Genesupport SA, Geneva, Switzerland). Each time before high-throughput NGS, the quality and pooling process were analyzed by MiSeq (preliminary run exploiting 50 bp). The libraries were sequenced by exploiting the 150 bp paired-end protocol. Parameters describing the quality of the cDNA libraries and NGS were presented in our previous paper [18]. The in vitro culture of the embryonic axes, RNA isolation, and NGS were performed with three independent replicates. Raw transcriptomics data have been deposited in the SRA database. The BioProject accession number for white lupin is PRJNA953600, https://www.ncbi.nlm.nih.gov/sra/PRJNA953600, deposited on 11 April 2023, and for Andean lupin, it is PRJNA953433, https://www.ncbi.nlm.nih.gov/sra/PRJNA953433, deposited on 9 April 2023.

The sequencing raw data after demultiplexing (splitting of the raw sequencing data into separate files using sample-specific barcodes) were cleaned by eliminating the adapter reads, N-base reads, and low-quality reads using the CLC Genomics Workbench trim sequences module (Qiagen version 20). For expressed gene analysis, the expression level of each gene in each library was calculated by quantifying the number of Illumina reads that mapped to the narrowleaf lupin (*Lupinus angustifolius* L.) transcriptome (LupAngTanjil_v1.0) reference sequence annotated with cDNAs (the number of contigs was 57,263) using the CLC Genomics Workbench RNA-Seq Analysis module. The raw gene expression counts were normalized using the RPKM method described by Mortazavi and coworkers [30]. The RPKM value represents the reads per kilobase of the exon model per million mapped reads.

The heatmaps were prepared from records selected on a subcellular localization basis and were made with the software Heatmapper http://www.heatmapper.ca, accessed on 26 March 2023, using the ‘normalized means’ from three independent transcriptome sequencings (Appendix A).

### 4.3. Prediction of Lupin Lipases’ Subcellular Localizations

Prediction of the lupin lipases’ subcellular localizations was performed with the software DeepLoc 2.0 https://services.healthtech.dtu.dk/services/DeepLoc-2.0/, accessed on 16 February 2023, using a high-quality model with a long output format.

### 4.4. Comparison of Selected Lupin Lipases’ Amino Acid Sequences with Yeast Atg15

Amino acid sequences of 38 lupin lipases were analyzed according to the Atg15 (QHB07198.1) yeast (*Saccharomyces cerevisiae*), which was used as a template. The phylogenetic tree was constructed using the neighbor-joining method. The similarity was analyzed using Job Dispatcher software (https://wwwdev.ebi.ac.uk/Tools/jdispatcher/, accessed on 26 March 2023).

The lupin lipases (XP_019420123.1, XP_019425758.1, and XP_019430451.1) most related to the Atg15 (QHB07198.1) yeast were found to have the IWVTGHSLGG amino acid sequence. The amino acid sequences of the selected lupin lipases were entered into MegAlign Pro (DNASTAR Lasergene) and aligned using the Clustal Omega algorithm.

## 5. Conclusions

During seed germination, lipases are the primary enzymes responsible for the initiation of storage lipid breakdown. They operate on lipid droplets deposited in the cytoplasm and release fatty acids from triacylglycerols. However, we formulated a hypothesis related to another important function of lipases in plants, namely, their involvement in the vacuole-localized degradation of autophagic bodies. An analysis of white and Andean lupin transcriptomes identified dozens of lipases, also with probable vacuolar localization. Additionally, some of them demonstrated similarities in amino acid sequence in the lipase active-site motif with the Atg15 yeast, which possesses lipolytic activity, and its involvement in autophagic body degradation is already well-confirmed. Taking into consideration also that under sugar-starvation conditions, lipolytic activity is elevated in lupin embryonic axes with simultaneously higher lipid contents than in non-starved axes, we postulated that an increase in lipolytic activity is linked with autophagy, which is also enhanced under sugar-starvation conditions and may be related to autophagic body degradation. Currently, our hypothesis requires verification, and further investigation is necessary to answer definitively whether lipases are involved in the degradation of autophagic bodies in plants.

## Figures and Tables

**Figure 1 ijms-25-00090-f001:**
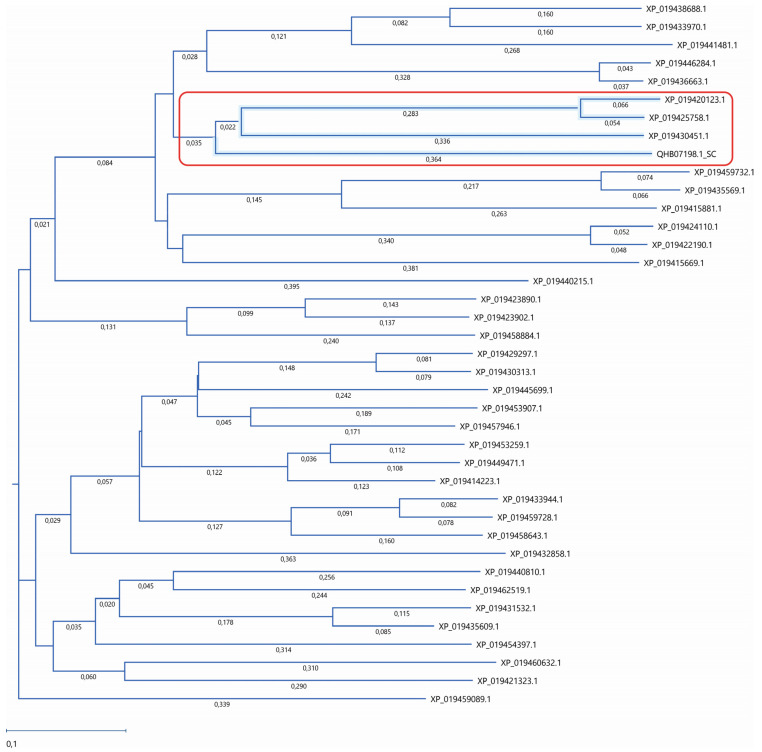
Phylogenetic analysis of lupin lipases. The amino acid sequences of 38 lupin lipases (listed in Table 1) were analyzed according to the Atg15 (QHB07198.1) yeast (*Saccharomyces cerevisiae*) as a template. The phylogenetic tree was constructed using the neighbor-joining method. The similarity was analyzed using Job Dispatcher software (https://wwwdev.ebi.ac.uk/Tools/jdispatcher/, accessed on 26 March 2023). The lupin lipases most related to Atg15 (QHB07198.1) are marked with a red rectangle.

**Figure 2 ijms-25-00090-f002:**
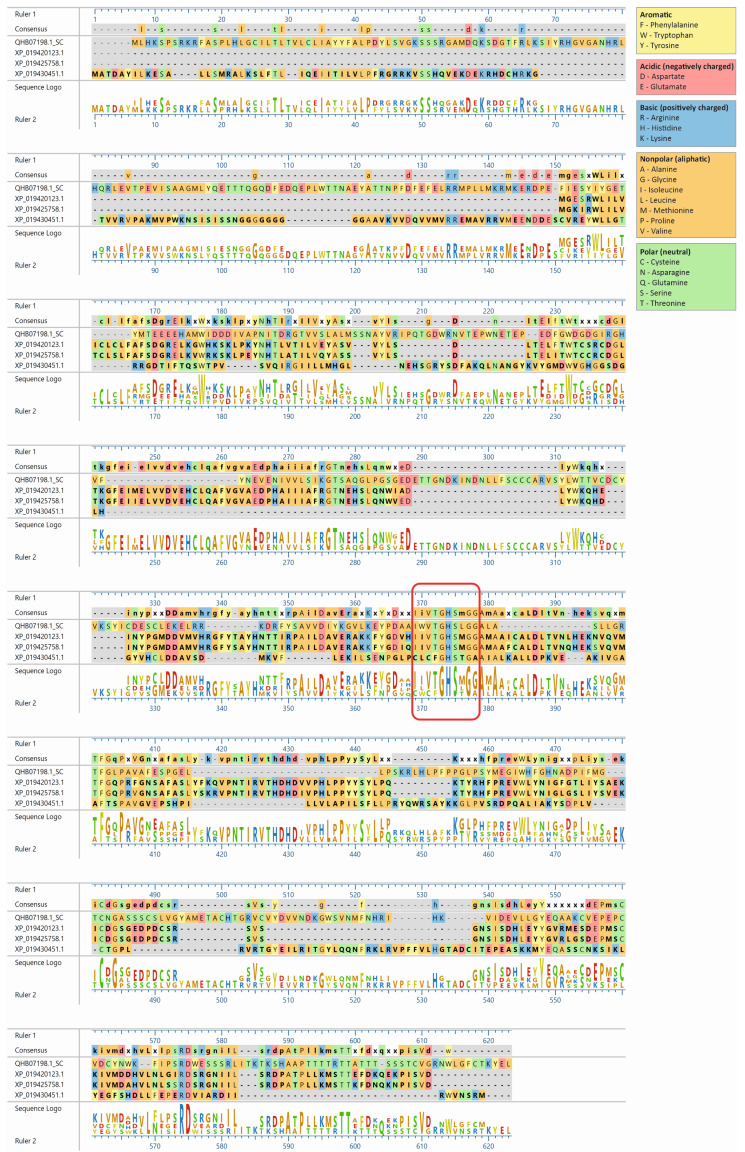
Amino acid sequence alignment of the three lupin lipases compared to the Atg15 (QHB07198.1) yeast (*Saccharomyces cerevisiae*). The amino acid sequences of the selected proteins were entered into MegAlign Pro (DNASTAR Lasergene) and aligned using the Clustal Omega algorithm. The amino acid sequence IWVTGHSLGG, a putative lipase active-site motif of Atg15 [11], is marked with a red rectangle.

**Figure 3 ijms-25-00090-f003:**
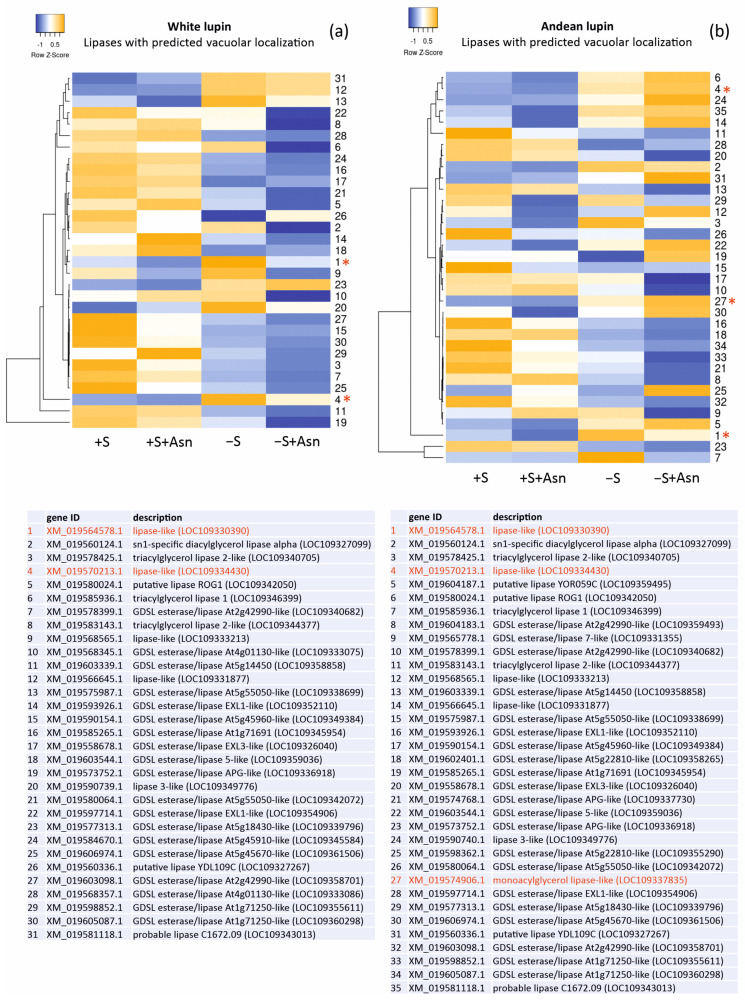
Heatmaps showing the relations in the levels of transcripts of the genes encoding lipases with the predicted vacuolar localization (Table 1) in the white lupin (**a**) and Andean lupin (**b**) embryonic axes cultured in vitro for 96 h on a medium with 60 mM of sucrose (+S) or without sugar (−S). The media were enriched with 35 mM of asparagine (+Asn). The asterisks and red fonts indicate the gene transcripts with close phylogenetic relations and amino acid sequence similarities to the Atg15 yeast (*Saccharomyces cerevisiae*) (Figure 1). The data represent the averages obtained from three independent experiments. The transcriptomics data selected for the preparation of this figure are presented in Appendix A.

**Table 1 ijms-25-00090-t001:** Prediction of the subcellular localization of the selected lipases (only lipases whose vacuolar localization probabilities were above the threshold value) identified in transcriptomes of the white and Andean lupin embryonic axes mapped to the narrowleaf lupin (*Lupinus angustifolius* L.) transcriptome (LupAngTanjil_v1.0). DeepLoc 2.0 software (https://services.healthtech.dtu.dk/services/DeepLoc-2.0/, accessed on 16 February 2023) with a high-quality model was used. A localization was predicted if its probability was above the threshold value (the green-filled cells), and the list was sorted by the highest score for vacuolar location. The red fonts indicate the lupin lipase proteins with the highest amino acid sequence similarity to the Atg15 (QHB07198.1) yeast (*Saccharomyces cerevisiae*). The similarity was analyzed using Job Dispatcher software (https://wwwdev.ebi.ac.uk/Tools/jdispatcher/, accessed on 26 March 2023), and the results of the analysis are presented in Figure 1. Data related to the predicted subcellular localization of all the lipases identified in the lupin embryonic axes are presented in Appendix A.

No.	Gene ID	Protein ID	Name	Cytoplasm	Nucleus	Extra-Cellular	Cell Membrane	Mitochondrion	Plastid	Endoplasmic Reticulum	Vacuole	Golgi Apparatus	Peroxisome	Lupin Species
			Probability threshold	0.4761	0.5014	0.6173	0.5646	0.6220	0.6395	0.6090	0.5848	0.6494	0.7364	
1.	XM_019564578.1	XP_019420123.1	lipase-like (LOC109330390)	0.1249	0.0637	0.6849	0.2738	0.0537	0.0589	0.4632	0.7413	0.4305	0.0051	white and Andean
2.	XM_019560124.1	XP_019415669.1	sn1-specific diacylglycerol lipase alpha (LOC109327099)	0.3913	0.62264	0.0683	0.5681	0.3879	0.1867	0.5224	0.7265	0.6729	0.0083	white and Andean
3.	XM_019578425.1	XP_019433970.1	triacylglycerol lipase 2-like (LOC109340705)	0.1372	0.0666	0.5950	0.2569	0.0253	0.2662	0.4241	0.7121	0.3615	0.0235	white and Andean
4.	XM_019570213.1	XP_019425758.1	lipase-like (LOC109334430)	0.1238	0.0633	0.7455	0.2846	0.0734	0.1256	0.4167	0.7070	0.4139	0.0128	white and Andean
5.	XM_019604187.1	XP_019459732.1	putative lipase YOR059C (LOC109359495)	0.5098	0.3743	0.1712	0.3106	0.2847	0.2233	0.6089	0.6995	0.6764	0.0244	Andean
6.	XM_019580024.1	XP_019435569.1	putative lipase ROG1 (LOC109342050)	0.4751	0.3420	0.1784	0.3196	0.3168	0.2231	0.5423	0.6956	0.6792	0.0250	white and Andean
7.	XM_019585936.1	XP_019441481.1	triacylglycerol lipase 1 (LOC109346399)	0.1717	0.1104	0.3269	0.2610	0.0743	0.1614	0.6899	0.6813	0.3701	0.0266	white and Andean
8	XM_019604183.1	XP_019459728.1	GDSL esterase/lipase At2g42990-like (LOC109359493)	0.1070	0.0679	0.5973	0.2432	0.0105	0.1666	0.4311	0.6752	0.2104	0.0049	Andean
9.	XM_019565778.1	XP_019421323.1	GDSL esterase/lipase 7-like (LOC109331355)	0.1720	0.0840	0.6110	0.2890	0.0218	0.0991	0.1960	0.6684	0.3304	0.0367	Andean
10.	XM_019578399.1	XP_019433944.1	GDSL esterase/lipase At2g42990-like (LOC109340682)	0.1053	0.0639	0.6134	0.2579	0.0107	0.1252	0.4020	0.6563	0.2294	0.0056	white and Andean
11.	XM_019583143.1	XP_019438688.1	triacylglycerol lipase 2-like (LOC109344377)	0.0996	0.0750	0.5750	0.2067	0.0248	0.3830	0.4561	0.6560	0.2174	0.0211	white and Andean
12.	XM_019568565.1	XP_019424110.1	lipase-like (LOC109333213)	0.4079	0.1664	0.2777	0.1933	0.3235	0.2462	0.7146	0.6556	0.7723	0.0595	white and Andean
13	XM_019568345.1	XP_019423890.1	GDSL esterase/lipase At4g01130-like (LOC109333075)	0.1253	0.0987	0.4851	0.2872	0.0131	0.0569	0.5135	0.6525	0.2822	0.0057	white
14.	XM_019603339.1	XP_019458884.1	GDSL esterase/lipase At5g14450 (LOC109358858)	0.1087	0.0985	0.7378	0.1742	0.0168	0.1280	0.3941	0.6491	0.1522	0.0055	white and Andean
15.	XM_019566645.1	XP_019422190.1	lipase-like (LOC109331877)	0.4394	0.1665	0.3044	0.2291	0.3315	0.1615	0.6810	0.6468	0.7454	0.0281	white and Andean
16	XM_019575987.1	XP_019431532.1	GDSL esterase/lipase At5g55050-like (LOC109338699)	0.1232	0.0934	0.6040	0.2375	0.0122	0.0618	0.4224	0.6425	0.2297	0.0072	white and Andean
17.	XM_019593926.1	XP_019449471.1	GDSL esterase/lipase EXL1-like (LOC109352110)	0.1276	0.0907	0.6176	0.2751	0.0192	0.0835	0.2316	0.6369	0.2284	0.0112	white and Andean
18.	XM_019590154.1	XP_019445699.1	GDSL esterase/lipase At5g45960-like (LOC109349384)	0.1051	0.1087	0.5512	0.2341	0.0115	0.1143	0.3511	0.6302	0.1974	0.0055	white and Andean
19.	XM_019602401.1	XP_019457946.1	GDSL esterase/lipase At5g22810-like (LOC109358265)	0.1301	0.0966	0.6737	0.2306	0.0147	0.0618	0.4086	0.6205	0.1480	0.0043	Andean
20.	XM_019585265.1	XP_019440810.1	GDSL esterase/lipase At1g71691 (LOC109345954)	0.1187	0.0839	0.6802	0.2930	0.0111	0.0466	0.2459	0.6196	0.1618	0.0064	white and Andean
21	XM_019558678.1	XP_019414223.1	GDSL esterase/lipase EXL3-like (LOC109326040)	0.1235	0.0754	0.5616	0.2526	0.0118	0.0480	0.5097	0.6192	0.2622	0.0051	white and Andean
22.	XM_019574768.1	XP_019430313.1	GDSL esterase/lipase APG-like (LOC109337730)	0.0857	0.0713	0.7163	0.2529	0.0099	0.1867	0.4058	0.6156	0.1377	0.0076	Andean
23.	XM_019603544.1	XP_019459089.1	GDSL esterase/lipase 5-like (LOC109359036)	0.0988	0.0939	0.5668	0.2812	0.0168	0.0886	0.3382	0.6152	0.2027	0.0060	white and Andean
24.	XM_019573752.1	XP_019429297.1	GDSL esterase/lipase APG-like (LOC109336918)	0.0973	0.0867	0.6546	0.2086	0.0087	0.1593	0.3978	0.6094	0.1387	0.0056	white and Andean
25.	XM_019590739.1	XP_019446284.1	lipase 3-like (LOC109349776)	0.3293	0.1463	0.1357	0.2716	0.2600	0.1613	0.6459	0.6063	0.6241	0.1617	white
	XM_019590740.1	XP_019446285.1												Andean
26.	XM_019598362.1	XP_019453907.1	GDSL esterase/lipase At5g22810-like (LOC109355290)	0.1068	0.0656	0.7004	0.1867	0.0132	0.1135	0.3575	0.6060	0.1499	0.0088	Andean
27.	XM_019580064.1	XP_019435609.1	GDSL esterase/lipase At5g55050-like (LOC109342072)	0.1154	0.1049	0.5881	0.2344	0.0153	0.0821	0.3895	0.6032	0.2036	0.0039	white and Andean
28.	XM_019574906.1	XP_019430451.1	monoacylglycerol lipase-like (LOC109337835)	0.1607	0.1250	0.4608	0.2318	0.0163	0.1849	0.2884	0.6006	0.2001	0.0119	Andean
29.	XM_019597714.1	XP_019453259.1	GDSL esterase/lipase EXL1-like (LOC109354906)	0.1607	0.1250	0.4608	0.2318	0.0163	0.1849	0.2884	0.6006	0.2001	0.0119	white and Andean
30.	XM_019577313.1	XP_019432858.1	GDSL esterase/lipase At5g18430-like (LOC109339796)	0.1155	0.1176	0.6524	0.1599	0.0163	0.2249	0.3856	0.5976	0.1010	0.0071	white and Andean
31.	XM_019584670.1	XP_019440215.1	GDSL esterase/lipase At5g45910-like (LOC109345584)	0.1155	0.1176	0.6524	0.1599	0.0163	0.2249	0.3856	0.5976	0.1010	0.0071	white
32.	XM_019606974.1	XP_019462519.1	GDSL esterase/lipase At5g45670-like (LOC109361506)	0.1745	0.1344	0.6873	0.2629	0.0219	0.0575	0.3940	0.5970	0.1796	0.0034	white and Andean
33.	XM_019560336.1	XP_019415881.1	putative lipase YDL109C (LOC109327267)	0.5177	0.2787	0.1145	0.5016	0.2160	0.3013	0.5523	0.5947	0.4328	0.0594	white and Andean
34.	XM_019603098.1	XP_019458643.1	GDSL esterase/lipase At2g42990-like (LOC109358701)	0.1742	0.1030	0.6854	0.1671	0.0073	0.1844	0.3105	0.5943	0.1800	0.0192	white and Andean
35.	XM_019568357.1	XP_019423902.1	GDSL esterase/lipase At4g01130-like (LOC109333086)	0.0970	0.0669	0.6555	0.2113	0.0174	0.0771	0.3975	0.5898	0.1388	0.0032	white
36.	XM_019598852.1	XP_019454397.1	GDSL esterase/lipase At1g71250-like (LOC109355611)	0.1263	0.1557	0.6132	0.2149	0.0178	0.1169	0.3113	0.5887	0.1588	0.0057	white and Andean
37.	XM_019605087.1	XP_019460632.1	GDSL esterase/lipase At1g71250-like (LOC109360298)	0.1523	0.1259	0.5000	0.3009	0.0281	0.1422	0.3990	0.5884	0.3172	0.0111	white and Andean
38.	XM_019581118.1	XP_019436663.1	probable lipase C1672.09 (LOC109343013)	0.2891	0.1719	0.1389	0.3496	0.2474	0.1869	0.6562	0.5852	0.6347	0.1633	white and Andean

## Data Availability

The transcriptomic data (NGS) have been deposited in the SRA database. The BioProject accession number for white lupin is PRJNA953600, https://www.ncbi.nlm.nih.gov/sra/PRJNA953600, deposited on 11 April 2023; and for Andean lupin, it is PRJNA953433, https://www.ncbi.nlm.nih.gov/sra/PRJNA953433, deposited on 9 April 2023.

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
