# Peer review of "Identification and Potential Participation of Lipases in Autophagic Body Degradation in Embryonic Axes of Lupin (Lupinus spp.) Germinating Seeds"

_ijms, 2023, doi:10.3390/ijms25010090_

Round 1
Reviewer 1 Report
Comments and Suggestions for Authors
Major points:
The identification of lipases involved in autophagic body degradation is an interesting and rewarding of research. However, it is completely insufficient for the authors in this article to have simply applied the transcriptome to analyze the transcript levels of lipase-related genes:
1) Is the choice of yeast Atg15 as a reference sufficiently scientific?
2) Application of qPCR to verify the transcript levels of relevant genes is necessary.
3) Subcellular locations of the three potential lipases also need to be identified, rather than simple bioinformatics analysis.
Minor points:
1) Line 72 and Line 246, ‘in vitro’ should be italic,
2) Table 1, What does the green background refer to? In addition, horizontal distribution of Table 1 might be better than vertical distribution.
Author Response
Reviewer 1
Major points:
The identification of lipases involved in autophagic body degradation is an interesting and rewarding of research. However, it is completely insufficient for the authors in this article to have simply applied the transcriptome to analyze the transcript levels of lipase-related genes:
1) Is the choice of yeast Atg15 as a reference sufficiently scientific?
2) Application of qPCR to verify the transcript levels of relevant genes is necessary.
3) Subcellular locations of the three potential lipases also need to be identified, rather than simple bioinformatics analysis.
We must emphasize the fact that our manuscript is a BRIEF REPORT but not a final and complete RESEARCH ARTICLE. Our main goal is to present our hypothesis about the involvement of lipases in the degradation of the autophagic body in plants. Therefore, in the manuscript, we present only results that constitute the basis for the hypothesis. The next step will be to falsify the hypothesis and then all the research mentioned above by the reviewer will certainly be performed. We are even performing and planning a much broader scope of research to gather evidence for the involvement of lipases in the degradation of autophagic bodies, and when we collect such evidence, only then will we prepare the final Research Article. Therefore, agreeing with the reviewer that the above-mentioned research is important and necessary, we do not want to expand the set of results in this manuscript, leaving it as a BRIEF REPORT in which we only present our hypothesis along with the necessary premises supporting it.
Why do we refer to yeast Atg15? Because yeast Atg15 is the only protein known to science that participates in the degradation of the autophagic body and is a lipase. There is no other protein with lipolytic activity that has been proven to be involved in the degradation of the autophagic body in any organism. Therefore, keeping additionally in mind the highly evolutionary conserved feature of autophagy and the similarity of plant and yeast autophagy, we do not see any option to choose another reference protein. I can only add in our defence, that we provided such information in a few places (Introduction, Results, and Discussion) of the original manuscript.
Minor points:
1) Line 72 and Line 246, ‘in vitro’ should be italic,
Corrected (line 24, 74, 265, and 275 of the revised manuscript).
2) Table 1, What does the green background refer to? In addition, horizontal distribution of Table 1 might be better than vertical distribution.
The green background of a cell in Table 1 indicates the score of the prediction of the subcellular localization that is above the threshold value. However, we would kindly like to point out that such information was in the heading of Table 1 in the original manuscript.
Why Table 1 is arranged vertically? We do not know, but we fully agree with the reviewer that it should be arranged horizontally. We can only inform you that we submitted the manuscript with a horizontally arranged table, but we do not know why reviewers received the manuscript with a table that is arranged vertically. We have restored the horizontal arrangement of the table in the revised manuscript.

Reviewer 2 Report
Comments and Suggestions for Authors
The manuscript is an attempt to identify markers (lipases) of autophagy in plants. The subject is of interest because the information concerning the role of lipases in plant autophagy is scarce. The title of the work should be modified to make it more consistent with the tests that were carried out. There is a lack of references in the introduction. A deeper discussion about the role of lipases on lipid storage degradation during germination must be included. More details about the material used for transcriptome analysis must be given. The scientific name of the species must be written in italic. These comments were also made in the uploaded manuscript.

I've made some corrections in the ms, but in general que quality of the English language is good.
Author Response
Reviewer 2
The manuscript is an attempt to identify markers (lipases) of autophagy in plants. The subject is of interest because the information concerning the role of lipases in plant autophagy is scarce.
We are thankful to the Reviewer for the confirmation of our point of view on the scare-described role of lipases in plant autophagy.
The title of the work should be modified to make it more consistent with the tests that were carried out.
The title is modified. We are thankful to the reviewer for the suggestion; however, we modified the title in another way, which linked our manuscript to autophagy, not only lipases.
There is a lack of references in the introduction.
Corrected. We added references in several places indicated by the reviewer. This caused changes in the reference numbering, but the changed numbering is not marked by the “Track Changes” function of MS Word because we used EndNote 20, and this software blocks the “Track Changes”.
A deeper discussion about the role of lipases on lipid storage degradation during germination must be included.
To respond to the reviewer’s recommendation we reorganized and slightly extended the comments in the second paragraph of the Discussion (lines 219-235 of the revised manuscript). However, we would like not to extend the discussion about the involvement of lipases in the degradation of storage lipid accumulated in lipid droplets because this topic is not the main object of our manuscript, and we are afraid that more extensive comment on the role of lipases in storage lipid breakdown may dilute the main message of our manuscript.
More details about the material used for transcriptome analysis must be given.
Added accordingly with the reviewer's suggestion; at the beginning of section 4.2.
The scientific name of the species must be written in italic.
Corrected (lines 298-299 of the revised manuscript).
These comments were also made in the uploaded manuscript.
All these comments were taken under consideration by us and detailed responses are below.
I've made some corrections in the ms, but in general que quality of the English language is good.
We are thankful to the Reviewer for a positive opinion regarding the English language. We can add, that the manuscript before submission underwent comprehensive editing made by a native speaker and bioscience editor Richard Ashcroft http://www.anglopolonia.com/. Hopefully, during the revision, especially introducing the corrections related to Reviewer 2 remarks, we have made additional corrections to improve the English language.
Responses to Reviewer 1 comments from the manuscript.
From the data presented, the authors can only conclude that lipases with some homology to those of yeast were detected in Lupinus. There is no physiological or biochemical evidence that they have the same role, so the title must be modified. I suggest the following title. Maybe germinations instead development (page 1 of the original manuscript)
We have changed the title; however, we modified the reviewer’s suggestion, because we would like to emphasize in the title, that the research is linked with autophagy but not only to lipases in plants.
Language corrections.
We introduced almost all language corrections following the reviewer’s suggestions throughout the whole manuscript.
Add a reference (Introduction)
Corrected. We added references in several places indicated by the reviewer. This caused changes in the reference numbering, but the changed numbering is not marked by the “Track Changes” function of MS Word because we used EndNote 20, and this software blocks the “Track Changes”.
reduce the number of ref (line 64 of the original manuscript)
Corrected – reduced to 4 references in this place (line 66 of the revised manuscript). We would like not to reduce the references too much in this place because 4 left references were published in 2021 and 2023, and it emphasizes the novelty of the literature data and our hypothesis.
The authors must give some information about the germination results. After 4d in the different media was the germination process identical for the axis of both species? Was there any effect of sucrose or asp on the germination process? (page 2 of the original manuscript)
We would like not to provide such data in this manuscript because results related to several morpho-physiological parameters of 4-day-old lupin embryonic axes were already published in our earlier papers. Also, in our opinion, such data is not strictly related to and necessary for our hypothesis presented in this manuscript. However, to respond to this reviewer's requirement we reorganized the beginning of the Results section, and we added in this place information about our two already published papers where many morpho-physiological parameters of 96-hour lupin embryonic axes are already described (lines 99-109).
Give here the values of similarity (page 6 of the original manuscript)
Done - values of the identity and similarity for three lupin lipases to Atg15 were added to the manuscript in this place (lines 134-136 of the revised manuscript).
add a ref (page 10, line 186 of the original manuscript)
Corrected – added (line 198 of the revised manuscript)
Lipases are also involved on the degradation of storage lipids that accumulate in oleosomes (oil bodies) during embryo germination. The possibility that some of the lipases found in this work could be more related with lipid degradation rather than autophagy must be deeply discussed. (page 10 of the original manuscript)
Additional comments related to the reviewer's suggestion we introduced in the second paragraph of the Discussion (lines 219-235). However, as we wrote above, we would like not to extend the discussion about the involvement of lipases in the degradation of storage lipid accumulated in lipid droplets because this topic is not the main object of our manuscript, and we are afraid that more extensive comment on the role of lipases in storage lipid breakdown may dilute the main message of our manuscript.
use lowercase letters. Check through the ms (page 13 of the original submission)
We checked and corrected the writing if necessary of all references. We used EndNote 20 for references thus we hope that references are without mistakes.

Round 2
Reviewer 1 Report
Comments and Suggestions for Authors
The author answered my query.
Reviewer 2 Report
Comments and Suggestions for Authors
The authors have changed the manuscripts according to the suggestions. Thus, in my opinion, the new version can be accepted for publication.
Comments on the Quality of English LanguageThe English language is OK.